# Coronavirus testing indicates transmission risk increases along wildlife supply chains for human consumption in Viet Nam, 2013-2014

Nguyen Quynh Huong[1☯], Nguyen Thi Thanh Nga[1☯], Nguyen Van Long[2], Bach Duc Luu[2], Alice Latinne[1,3,4], Mathieu Pruvot[3], Nguyen Thanh Phuong[5], Le Tin Vinh Quang[5], Vo Van Hung[5], Nguyen Thi Lan[6], Nguyen Thi Hoa[6], Phan Quang Minh[2], Nguyen Thi Diep[2], Nguyen Tung[2¤a], Van Dang Ky[2¤a], Scott I. Roberton[1], Hoang Bich Thuy[1], Nguyen Van Long[1], Martin Gilbert[3¤b], Leanne Wicker[1¤c], Jonna A. K. Mazet[7], Christine Kreuder Johnson[7], Tracey Goldstein[7], Alex Tremeau-Bravard[7], Victoria Ontiveros[7], Damien O. Joly[3¤d], Chris Walzer[3,8], Amanda E. Fine[1,3‡]*, Sarah H. Olson[3‡]

1 Wildlife Conservation Society, Viet Nam Country Program, Ha Noi, Viet Nam, 2 Department of Animal Health, Ministry of Agricultural and Rural Development of Viet Nam, Ha Noi, Viet Nam, 3 Wildlife Conservation Society, Health Program, Bronx, New York, United States of America, 4 EcoHealth Alliance, New York, New York, United States of America, 5 Regional Animal Health Office No. 6, Ho Chi Minh City, Viet Nam, 6 Faculty of Veterinary Medicine, Viet Nam National University of Agriculture, Ha Noi, Viet Nam, 7 One Health Institute, School of Veterinary Medicine, University of California, Davis, California, United States of America, 8 Research Institute of Wildlife Ecology, University of Veterinary Medicine, Vienna, Austria

☯ These authors contributed equally to this work.
¤a Current address: The Animal Asia Foundation Viet Nam, Ha Noi, Viet Nam
¤b Current address: Cornell Wildlife Health Center, College of Veterinary Medicine, Cornell University, Ithaca, New York, United States of America
¤c Current address: Australian Wildlife Health Centre, Healesville Sanctuary, Zoos Victoria, Healesville, Victoria, Australia
¤d Current address: British Columbia Ministry of Environment and Climate Change Strategy, Victoria, British Columbia, Canada
‡ These authors also contributed equally to this work.
* afine@wcs.org

**Data Availability Statement:** All relevant data are available at https://doi.org/10.5061/dryad.7h44j0zrj.

## Abstract

Outbreaks of emerging coronaviruses in the past two decades and the current pandemic of a novel coronavirus (SARS-CoV-2) that emerged in China highlight the importance of this viral family as a zoonotic public health threat. To gain a better understanding of coronavirus presence and diversity in wildlife at wildlife-human interfaces in three southern provinces in Viet Nam 2013–2014, we used consensus Polymerase Chain Reactions to detect coronavirus sequences. In comparison to previous studies, we observed high proportions of positive samples among field rats (34.0%, 239/702) destined for human consumption and insectivorous bats in guano farms (74.8%, 234/313) adjacent to human dwellings. Most notably among field rats, the odds of coronavirus RNA detection significantly increased along the supply chain from field rats sold by traders (reference group; 20.7% positivity, 39/188) by a factor of 2.2 for field rats sold in large markets (32.0%, 116/363) and 10.0 for field rats sold and served in restaurants (55.6%, 84/151). Coronaviruses were also detected in rodents on the majority of wildlife farms sampled (60.7%, 17/28). These coronaviruses were found in the Malayan porcupines (6.0%, 20/331) and bamboo rats (6.3%, 6/96) that are raised on wildlife farms for human consumption as food. We identified six known coronaviruses in

**Funding:** This study was made possible by the generous support of the American people through the United States Agency for International Development (USAID) Emerging Pandemic Threats PREDICT project (cooperative agreement numbers GHN-A-OO-09-00010-00 [J.A.K.M., C.K.J., T.G., D. O.J.] and AID-OAA-A-14-00102 [J.A.K.M, C.K.J., T. G., A.E.F, S.H.O.]). The funders had no role in study design, data collection and analysis, decision to publish, or preparation of the manuscript. The URL to the USAID Emerging Pandemic Threats Program (EPT-1 and 2) is https://www.usaid.gov/ept2.

**Competing interests:** The authors have declared that no competing interests exist.

bats and rodents, clustered in three *Coronaviridae* genera, including the *Alpha*-, *Beta*-, and *Gammacoronaviruses*. Our analysis also suggested either mixing of animal excreta in the environment or interspecies transmission of coronaviruses, as both bat and avian coronaviruses were detected in rodent feces on wildlife farms. The mixing of multiple coronaviruses, and their apparent amplification along the wildlife supply chain into restaurants, suggests maximal risk for end consumers and likely underpins the mechanisms of zoonotic spillover to people.

## Introduction

Human-wildlife contact with a bat or an intermediate host species in China likely triggered a coronavirus spillover event that may have involved wildlife markets and led to the pandemic spread of SARS-CoV-2 [1,2]. The pandemic risk of commercial trade in live wildlife was first recognized during the 2002–2003 Severe Acute Respiratory Syndrome (SARS) outbreak due to SARS-CoV [3]. This virus spread to countries in Asia, Europe, and the Americas with 8,096 people infected and 774 deaths, costing the global economy about $US 40 billion in response and control measures [4,5]. Unfortunately, the impact of COVID-19, the disease caused by SARS-CoV-2, has reached nearly every country and greatly surpassed those numbers by many orders of magnitude [6]. While bats are thought to be the ancestral hosts for all groups of coronaviruses [7], for both SARS-CoV and SARS-CoV-2 wildlife trade supply chains are suspected to have contributed the additional conditions necessary for the emergence, spillover, and amplification of these viruses in humans [8,9]. In Viet Nam, between 2013 to 2014, we conducted coronavirus surveillance to understand the presence and diversity of coronaviruses in wildlife at sites identified as high-risk interfaces for viral spillover from wildlife to humans [10]. We sampled at three sub-interfaces along the live field rat trade (*Rattus* sp. and *Bandicota* sp.) including field rats sold by rat traders, by vendors in large markets, and rats butchered and sold in restaurants as prepared dishes. We also sampled rodents raised on wildlife farms to assess risk from different wildlife supply chains destined for human consumption. We sampled bat guano, primarily on bat guano farms to assess the potential occupational risk of this practice given that bat guano farm artificial roost structures are often erected near human dwellings.

In the early 2000s, the Vietnamese field rat trade was estimated to process 3,300–3,600 tons of live rats annually for consumption, a market valued at US$2 million [11]. Although rats are still commonly traded in wet markets and sold live for food consumption along the Mekong Delta in southern Viet Nam, no recent published data on the scale and scope of the trade is available [12]. This human-wildlife interface involves the capture of wild free-ranging field rats, subsequent trade, and consumption along a supply chain involving the entire Mekong Delta region, particularly Cambodia and Viet Nam [13]. Driving this trade are consumers in Viet Nam and Cambodia, some of whom report eating rats at least once per week because of their good flavor, low cost, and perception of rats as 'healthy, nutritious, natural, or disease free' [13]. Rat parts (heads, tails, and internal organs discarded at slaughter) are also often fed to domestic livestock or herptiles raised in captivity including frogs, snakes, and crocodiles [12].

Over the past three decades, commercial wildlife farming has developed in many countries in Southeast Asia, including Viet Nam. Although there are historic references to the occurrence of wildlife farms in Viet Nam dating back to the late 1800s, the rapid expansion in terms

of farm numbers, species diversity, and scale of operations has occurred in recent decades in response to growing domestic and international demand for wildlife [14]. A 2014 survey across 12 provinces in southern Viet Nam identified 6,006 registered wildlife farms of which 4,099 had active operations. The surveyed farms were stocked with approximately one million wild animals including, rodents, primates, civets, wild boar, Oriental rat-snakes, deer, crocodiles, and softshell turtles. Ninety-five percent of the farms held 1–2 species of wildlife, and 70% of the farms also raised domestic animals on the same premises [15]. A key component of the wildlife farm industry in Viet Nam is the raising of wild species for meat for human consumption [15]. These farms sell to urban wild meat restaurants serving increasingly affluent populations throughout the country and also supply international markets with wild meat [16]. Commercial wildlife farming in Viet Nam is part of the expanded international trade of wildlife that has been hypothesized to contribute to the cause of global epidemics, such as SARS [17] and now COVID-19.

Emerging evidence suggests zoonotic virus spillover risk is a concern at bat-human interfaces in Asia. Guano harvested from a cave in Thailand were positive for a group C betacoronavirus, which includes MERS-CoV, and 2.7% of 218 people living in close proximity to bats known to carry viruses related to SARS-CoV tested positive for SARS-related antibodies in China [18,19]. The traditional practice of guano farming in parts of Cambodia and Viet Nam involves the construction of artificial bat roosts in gardens or backyard farms, under which domestic animals and crops are raised, and children often play [20,21]. Cambodian development programs promoted the practice in 2004 to enhance soil fertility, reduce reliance on chemical fertilizers, generate income ($US 0.50/kg), control insect pests, and protect the lesser Asiatic yellow bats (*Scotophilus kuhlii*) that were being hunted [20–22]. No personal protection measures are taken when harvesting the guano, which is reported to improve the growth rate in five economically important plant species [23].

In this study we investigated the presence and diversity of coronavirus sequences in the field rat trade distribution chain, wildlife farms specializing in raising rodents for human consumption, and bat guano "farms" and roosts near human dwellings to better understand the natural hosts of coronaviruses and the risk for these interfaces to facilitate spillover into humans.

## Materials and methods

### Sampling locations

Sampling was performed at multiple sites representing several high-risk interfaces for contacts among people, rodents, and bats. Rodent sampling focused on the live field rat trade supply chain and wildlife farms specializing in raising rodents [Malayan porcupines (*Hystrix brachyura*) and bamboo rats (*Rhizomys* sp.)] for meat. Along the field rat supply chain, we targeted eight sites involved in the private sale and butchering of rats for consumption, defined as 'traders' for the purpose of this study in Dong Thap and Soc Trang provinces, 14 large market sites where rats were butchered and sold in Dong Thap and Soc Trang provinces (>20 vendors), and two restaurant sites in Soc Trang province where live rats were kept on the premises and butchered and served as food (Fig 1). The 28 rodent farm sites targeted in Dong Nai province produced Malayan porcupines and bamboo rats for human consumption (Fig 2). Other species observed or raised at the wildlife farm sites included dogs, cattle, pigs, chickens, ducks, pigeons, geese, common pheasant, monitor lizards, wild boar, fish, python, crocodiles, deer, civets, non-human primates as pets or part of private collections, free-flying wild birds, and free-ranging peri-domestic rats.

Bat sampling occurred at bat guano "farms" and a natural bat roost located at a religious site. Bat guano farms consisted of artificial roosts constructed with a concrete base and pillars

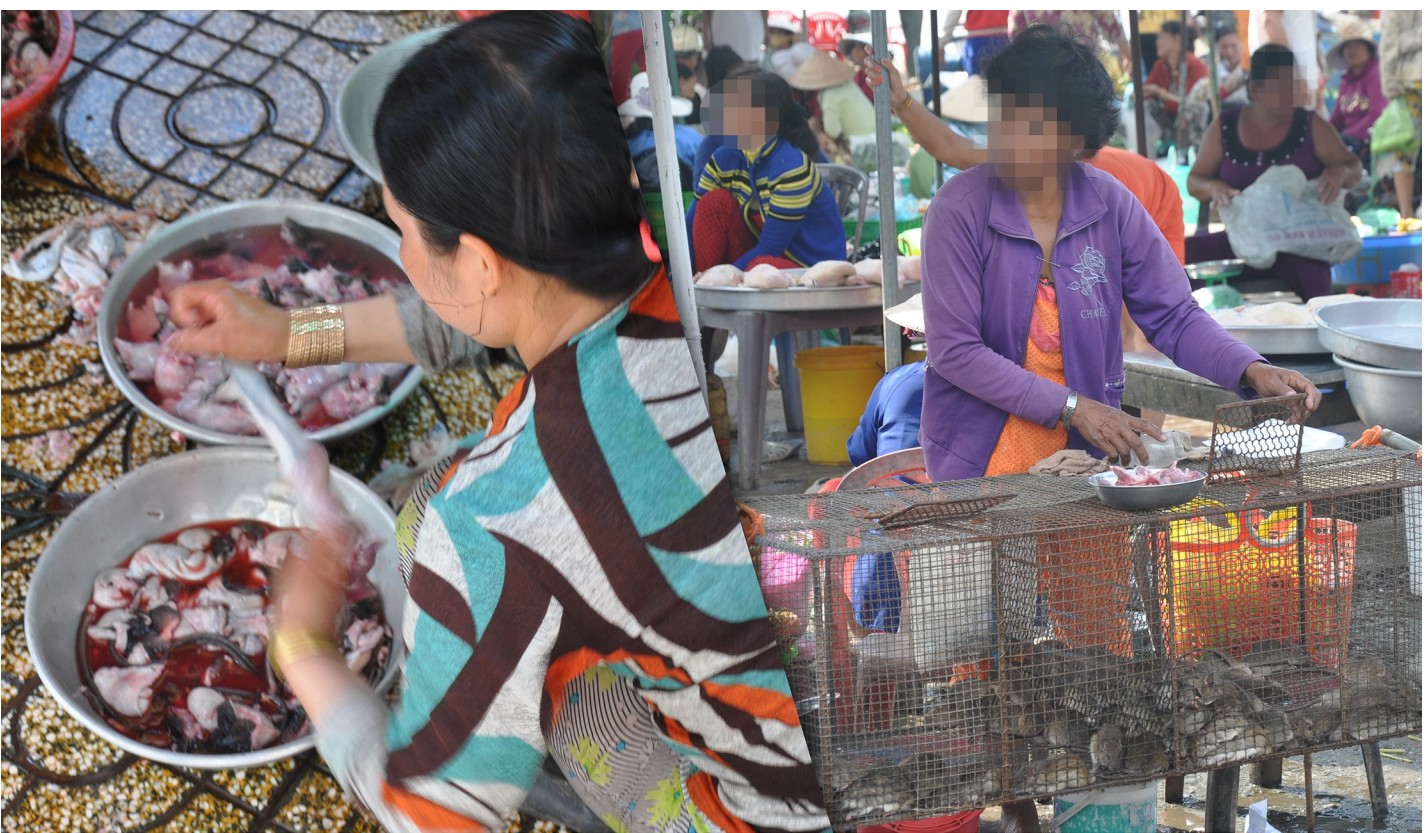

**Fig 1.** Slaughtering rats at a large market (left) and a rat vendor stall displaying live rats in cages in a large market (right) in Dong Thap province, October 2013.

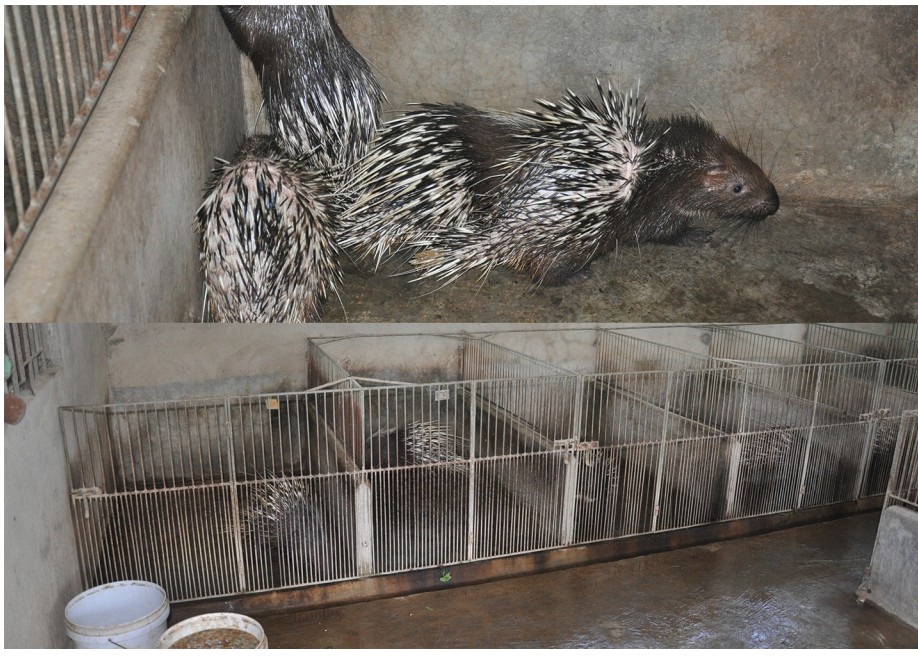

**Fig 2. Malayan porcupine (*Hystrix brachyura*) farm in Dong Nai province, November 2013.**

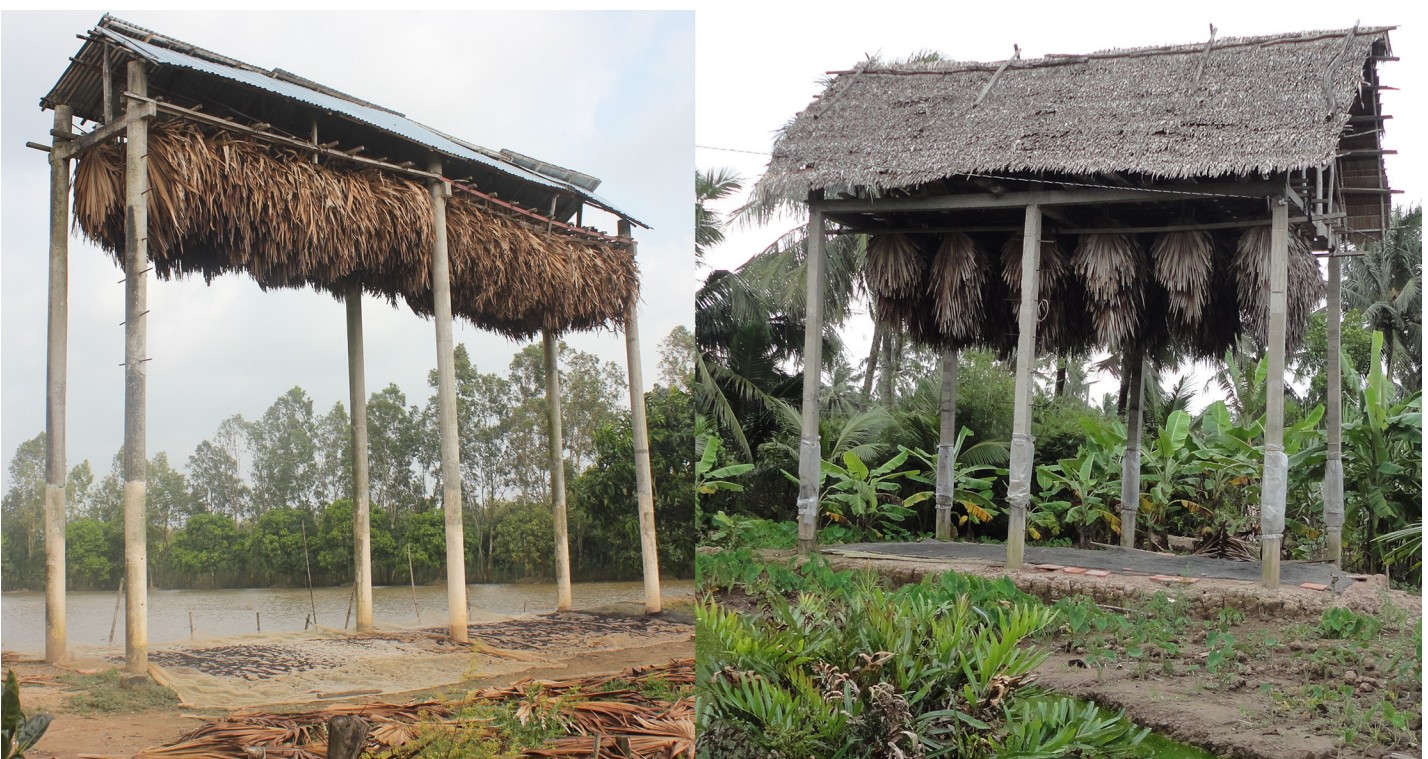

**Fig 3. Bat guano farms in Soc Trang Province, October 2013.**

topped with fronds of coconut palm or Asian Palmyra Palm (*Borassus flabellifer*) (Fig 3). Seventeen bat guano farms were sampled in the two provinces of Dong Thap and Soc Trang. The natural bat roost was located at a religious site in Soc Trang province known as the "bat pagoda", where *Pteropus* sp. have historically roosted in trees protected from hunting, and light and noise pollution [24].

All study sampling occurred from January 2013 to March 2014 at 41 sites in the wet (south Viet Nam: May 1st—November 30th) and 30 in the dry (south Viet Nam: December 1st—April 30th) seasons. Given the distances between sites, 69 sites were sampled once except the bat pagoda natural roost in Soc Trang province, which was visited three times and sampled in both seasons.

## Animal sampling

Samples from animals were humanely collected using standard and previously published protocols and no animals were purchased for this study (S1 Table) [25]. Samples from rats at all three sub-interfaces in the field rat trade were collected from individual carcasses after the rats were slaughtered by a trader, market vendor, or restaurant kitchen staff as part of the rat meat preparation process during normal sales to customers. Oral swabs were collected from the severed heads of all the rats, with at least one additional tissue sample collected from the discarded internal organs of each individual. The small intestine was the additional tissue most frequently collected with brain, kidney, lung, rectal swab, and urine also collected from some individuals. Rats were usually butchered at a common site for each observed time period that was only cleaned intermittently following the trader's, vendor's, or restaurant's regular practices.

Feces, urine, and swabs of the pen floors (environmental samples), were collected non-invasively (without handling the animals) from rodents on wildlife farms. Samples were classified as 'fecal sample' or 'urine sample' when voided feces or urine was collected from an animal housed individually in its own cage, and as 'environmental sample' when collected as a swab from cages housing multiple individuals.

Fecal samples and a small number of urine samples excreted by bats in guano farms and the natural roost site were collected on clean plastic cover sheets within 1–2 hours after placement under bat roosts, and thus each sample may represent one or multiple bats. Oral and rectal swabs were also collected from live-captured bats during one sampling visit at the natural pagoda roost site.

All animals were identified in the field to the lowest taxonomic level possible based on morphological characteristics, and species was identified in a subset of animals through genetic barcoding [15]. Due to difficulty of morphologic identification in the field, unless barcoded, rodents (*Rattus argentiventer*, *R. tanezumi*, *R. norvegicus*, *R. exulans*, *R. losea*, and *Bandicota indica*; [12,26]) were categorized as "field rats". Bats were classified as "*Microchiroptera*" following the traditional taxonomic classification (new classification of two new suborders *Yangochiroptera* and *Yinpterochiroptera*, was only published near the end of the study, so for consistency we used the historical classification [27]).

All samples were collected in cryotubes containing RNAlater (RNA stabilization reagent, Qiagen), and stored in liquid nitrogen in the field before being transported to the laboratory for storage at -80 ˚C. Samples were tested by the Regional Animal Health Office No. 6 (RAHO6) laboratory in Ho Chi Minh City. The study and sampling activities for specified dates and locations were approved by the Department of Animal Health of the Ministry of Agriculture and Rural Development and animal sampling protocols were reviewed by the Institutional Animal Care and Use Committee at the University of California at Davis (protocol number 16048).

## Sample testing

RNA was extracted (RNA MiniPrep Kit, Sigma-Aldrich) and cDNA transcribed (SuperScript III First Strand cDNA Synthesis System, Invitrogen). Coronavirus RNA was detected using two broadly reactive consensus nested-PCR assays targeting the *RNA dependent RNA polymerase (RdRp)* gene [28,29]. The positive control was a synthetic plasmid containing the primer-binding sites for both assays. Distilled water was used as a negative control and included in each test batch. PCR products were visualized using 1.5% agarose gels, and bands of the correct size were excised, cloned, and sequenced by Sanger dideoxy sequencing using the same primers as for amplification.

## Phylogenetic analysis

For sequence analysis and classification operating taxonomic units were defined with a cut off of 90% identity, i.e. virus sequences that shared less than 90% identity to a known sequence were labelled sequentially as PREDICT_CoV-1, -2, -3, etc. and groups sharing ≥ 90% identity to a sequence already in GenBank were given the same name as the matching sequence [7]. A phylogenetic tree was constructed for sequences amplified using the Watanabe protocol, as this PCR protocol yielded longer sequences and more positive results than the Quan protocol. Several representative sequences for each viral species found in our study were included for analysis and are available in GenBank (S3 Table). Alignments were performed using MUSCLE, and trees were constructed using Maximum likelihood and the Tamura 3-parameter model in MEGA7 [30]. The best-fit model of DNA substitution was selected in MEGA7 using BIC

scores (Bayesian Information Criterion) and Maximum Likelihood values (*lnL*). Bootstrap values were calculated after 1,000 replicates. In addition, a median-joining network was constructed using Network 5.0.0.3 [31] to explore phylogenetic relationships among bat coronavirus 512/2005 sequences at the intraspecies level, as haplotype networks may better represent the relationships among viral sequences with low sequence diversity compared with phylogenetic trees [32].

## Statistical analyses

Visualization of sampling locations in provinces in Viet Nam, along with the distribution by species and interface was constructed with the ggplot2 and sf packages [33]. The source of the Viet Nam provincial map is geoBoundaries v. 3.0.0 (https://www.geoboundaries.org; [34]) and Open Development Mekong (https://vietnam.opendevelopmentmekong.net). All analyses were done using R version 3.5.0 or higher (R Development Core Team, Vienna, Austria). Data (S1 Data) and code (S1 R Code) are available in the supplementary materials. The effect of risk factors (season, sub-interface type) was examined and limited to interfaces for which the distribution of samples across factors could support the analysis. These included season for *Pteropus* bat samples collected in the bat pagoda natural roost and the effect of season and sub-interface for samples collected in the rodent trade in southern Viet Nam. Given the low sample size, the effect of season for *Pteropus* bats samples positive for coronaviruses was assessed using a Fisher exact test. The effect of season (dry, wet, with dry season as reference category) and sub-interface type (trader, large markets, restaurants, with trader as reference category) in traded rodent samples positive for coronaviruses was assessed with a mixed effect multivariable logistic regression, with sites as random effect (i.e. grouping variable) using the lme4 R package [35]. A p-value of less than 0.05 was considered statistically significant. The 95% binomial confidence intervals for proportions were calculated using binom.test in R.

The comparison of the proportion of coronavirus positives in different sample types was performed on positive individuals sampled in the live field rat trade with multiple sample types collected per individual. We then calculated the proportion of individuals positive for each sample type, as a proxy for the probability of detection by each sample type.

## Results

### Detection of coronavirus by animal taxa and interface

A total of 2,164 samples collected between January 2013 and March 2014 from rodents and bats were tested for coronaviruses (Table 1, S1 Table). Assuming that non-invasive samples from bats and farmed rodents represented unique distinct individuals, these samples came from 1,506 individuals, including 1,131 rodents (702 field rats and 429 wildlife farm rodents) and 375 bats from 70 sites sampled in Dong Thap, Soc Trang, and Dong Nai provinces in the southern region near the Mekong River Delta (Fig 4).

Out of 70 sites, coronavirus positives were detected at 58 including 100% (24/24) of live rat trade sites, 60.7% (17/28) of rodent wildlife farm sites, 94.1% (16/17) of bat guano farm sites, and at the one natural pteropid bat roost. Wildlife farms were only sampled in Dong Nai province and the live rat trade and bat interfaces were sampled in Dong Thap and Soc Trang provinces (Fig 4).

Coronaviruses were detected in the field rat trade (a mix of *Rattus* and *Bandicota* genera) at all sites in Dong Thap (n = 16) and Soc Trang (n = 8) provinces, with 34.6% (95% CI 29.8–39.7%, 129/373) and 33.4% (95% CI 28.4–38.9%, 110/329) positives respectively. The overall proportion of positives in field rats was 34.0% (95% CI 30.6–37.7%, 239/702), ranging from 3.2% to 74.4% across sites. Field rats sampled at sub-interfaces in the live rat trade had an

**Table 1. Summary of coronavirus positives by taxa and interface.** Co-infection is defined as the detection of two different coronavirus taxonomic units in an individual animal.

| Taxa group | Interface | Sub-interface | Taxa group | % site positive | % individual positive | Viral species | # of co-infected animals |
|---|---|---|---|---|---|---|---|
| Rodents | Rat trade | Rat trader (selling live rats and slaughtering live rats for sale as meat) | Field rat[a] | 100% (8/8) | 20.7% (39/188) | Murine coronavirus (n = 36), Longquan aa coronavirus (n = 5) | 2 |
| | | Large market (selling live rats and slaughtering live rats for sale as meat) | Field rat[a] | 100% (14/14) | 32.0% (116/363) | Murine coronavirus (n = 103), Longquan aa coronavirus (n = 31) | 18 |
| | | Restaurant (slaughtering live rats held on the premises and preparing as food) | Field rat[a] | 100% (2/2) | 55.6% (84/151) | Murine coronavirus (n = 70), Longquan aa coronavirus (n = 20) | 6 |
| | Wildlife farm | | *Hystrix* sp. | 47.8% (11/23) | 6.0% (20/331) | Bat coronavirus 512/2005 (n = 19), Infectious bronchitis virus (IBV) (n = 1) | 0 |
| | | | *Rhizomys* sp. | 45.5% (5/11) | 6.3% (6/96) | Bat coronavirus 512/2005 (n = 5), Infectious bronchitis virus (IBV) (n = 1) | 0 |
| | | | *Rattus* sp.[b] | 100% (1/1) | 100% (1/1) | Bat coronavirus 512/2005 (n = 1) | 0 |
| | | | *Sciuridae* sp. | 0% (0/1) | 0% (0/1) | | |
| Bats | Human dwelling | Natural bat roost | | | | | |
| | | | *Pteropus* sp. | 100% (1/1) | 6.7% (4/60) | PREDICT_CoV-17 (n = 3), PREDICT_CoV-35 (n = 1) | 0 |
| | | | *Cynopterus horsfieldii* | 0% (0/1) | 0% (0/2) | | |
| | | Bat guano farm (artificial bat roost) | *Microchiroptera*[c] | 94.1% (16/17) | 74.8% (234/313) | PREDICT_CoV-17 (n = 1), PREDICT_CoV-35 (n = 38), Bat coronavirus 512/2005 (n = 216) | 21[d] |
| | | | | **82.9% (58/70)** | **33.5% (504/1506)** | | **47** |

[a] Field rat here refers to a mix of *Rattus* sp. and *Bandicota* sp.

[b] This environmental sample collected from a porcupine cage on a porcupine farm was barcoded as *Rattus* sp., suggesting this species was free-ranging at the site (Fig 2). The detection of a bat virus from this sample is suggestive of either environmental mixing or viral sharing.

[c] Suborder

[d] Co-infections included PREDICT_CoV-17 with Bat coronavirus 512/2005 (n = 1) and PREDICT_CoV-35 with Bat coronavirus 512/2005 (n = 20).

increasing proportion of positives along the distribution chain. Starting with traders, the proportion positive was 20.7% (95% CI 15.3–27.4%, 39/188), 32.0% (95% CI 27.2–37.1%, 116/363) in large markets, and 55.6% (95% CI 47.3–63.6%, 84/151) at restaurants (Fig 5). The proportion of positives was higher in the wet season (36.7%, 95% CI 32.8–40.8%, 210/572) than the dry season (22.3%, 95% CI 15.7–30.6%, 29/130). In a multivariate model with site as random effect, both season and sub-interface type were significantly associated with the risk of rat coronavirus infection, with higher risk of infection in the wet season (OR = 4.9, 95% CI 1.4–18.0), and increasing risk along the supply chain from traders (baseline) to large markets (OR = 2.2, 95% CI 1.05–4.7), to restaurants (OR = 10.0, 95% CI 2.7–39.5) (S2 Table). It should be noted, however, that since sites were only visited during one season, both independent variables were defined at the site level and confounding effects with other site-level characteristics cannot be excluded.

Among the positive field rats with more than one sample tested (n = 220), the proportion positive by sample type was 79.9% (95% CI 73.9–84.9%, 175/219) in oral swabs, 52.9% (95% CI

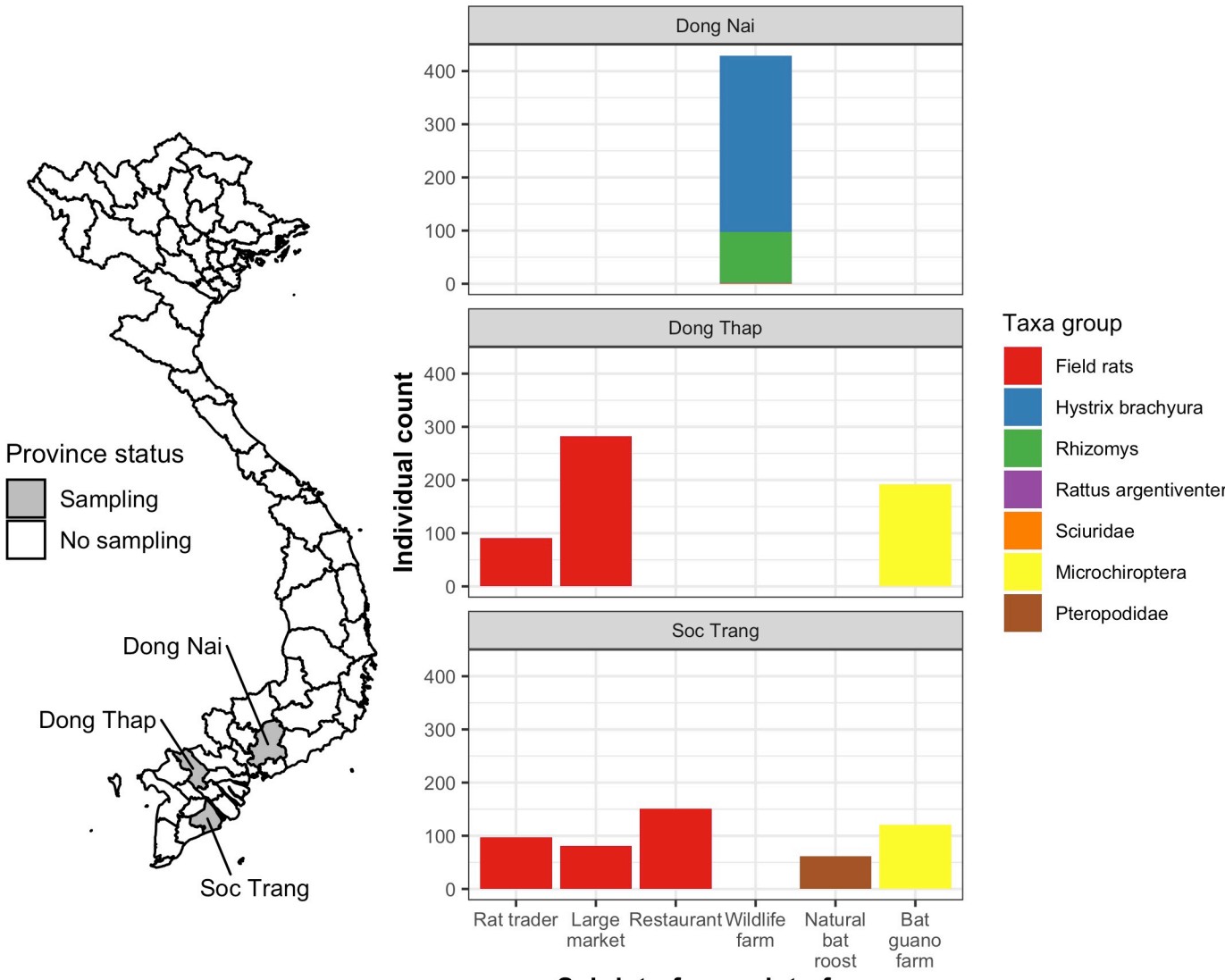

**Fig 4. Map of sampling sites by province and multi-panel plots showing individual counts of animals sampled by province, taxa, and sub-interface (rat trade) or interface.** The color of each bar represents the animal taxonomic group sampled in Dong Nai, Dong Thap, and Soc Trang provinces. *Sciuridae* and *Rattus argentiventer* were only sampled one time apiece from wildlife farms. Map was made using geoBoundaries v. 3.0.0 (https://www.geoboundaries.org; [34]) and Open Development Mekong (https://vietnam.opendevelopmentmekong.net) data under a CC BY 4.0 license.

38.6–66.8%, 27/51) in lung, 51.6% (95% CI 43.5–59.7%, 80/155) in small intestine, 31.2% (95% CI 12.1–58.5%, 5/16) in brain, 23.1% (95% CI 6.2–54.0%, 3/13) in kidney, 50.0% in feces (1/2), 100% in spleen (1/1), and 0% in urine/urogenital swabs (0/1).

At the rodent wildlife farm interface, 6.0% (95% CI 3.8–9.3%, 20/331) of *Hystrix brachyura* and 6.3% (95% CI 2.6–13.6%, 6/96) of *Rhizomys* sp. were positive. The overall proportion of positives was 6.3% (95% CI 4.3–9.1%, 27/429) (Table 1 and Fig 4). There was no difference among species or season and proportion positive in rodent farms, and low sample size and unequal sampling limited analysis.

The proportion of coronavirus positives at the two bat interfaces differed by an order of magnitude as 74.8% (95% CI 69.5–79.4%) of the non-invasive samples collected from *Microchiroptera* bats at bat guano farms were positive, and 6.7% (95% CI 2.2–17.0%) of the *Pteropus*

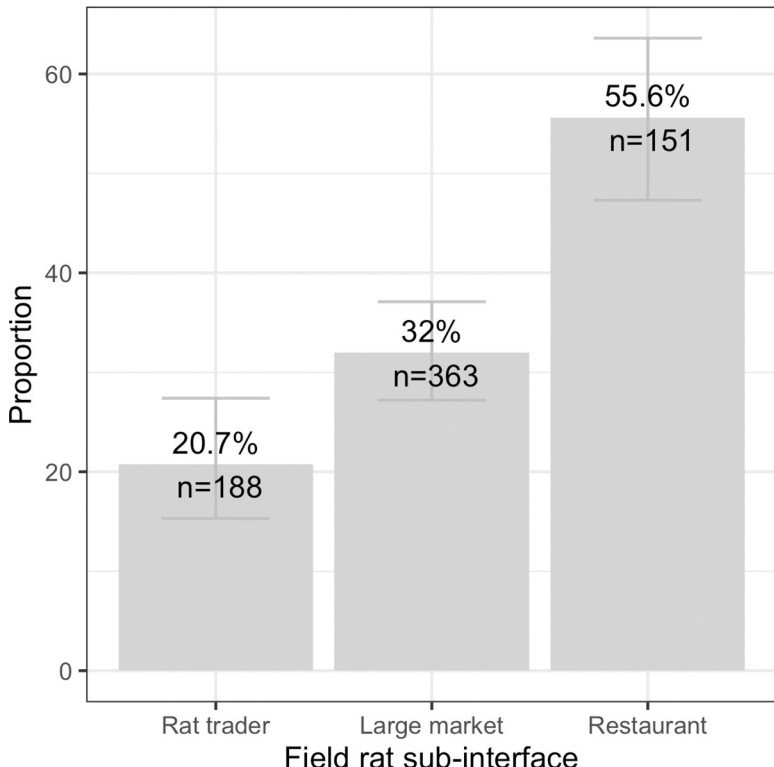

**Fig 5. Plot of the proportion of coronavirus positives in field rats by sub-interface in the live field rat trade chain.** Bars show 95% confidence intervals.

genus samples at the natural roost in Soc Trang province (Fig 4) were positive (Table 1). Pteropid bats sampled at the natural roost had higher proportions of positives in the wet season (27.3%, 95% CI 7.3–60.7%, 3/11) compared with the dry season (2.0%, 95% CI 0.1–12.2%, 1/ 50; Fisher exact test p = 0.02, OR = 16.6 [1.2–956.8]), although low sample size and single sampling per season warrants cautious interpretation.

## Phylogenetic analysis

Six distinct taxonomic units of coronaviruses corresponding to bat coronavirus 512/2005, Longquan aa coronavirus, avian infectious bronchitis virus (IBV), murine coronavirus, PRE-DICT_CoV-17, and PREDICT_CoV-35 were detected. All these viruses were detected using both the Watanabe and Quan assays, except IBV sequences that were detected only using the Quan protocol. Of the 504 positive animals, 433 were positive by the Watanabe assay, 410 were positive by the Quan assay, and 339 were positive by both. Phylogenetic analysis showed that among the six coronaviruses detected, PREDICT_CoV-35 and bat CoV 512/2005 clustered within the *Alphacoronaviruses*, while PREDICT_CoV-17, Longquan aa CoV and murine CoV clustered within the *Betacoronaviruses*. The virus identified within the *Gammacoronavirus* genus was avian IBV.

PREDICT_CoV-17 and PREDICT_CoV-35 were first reported by Anthony et al. [17]. We found PREDICT_CoV-17 in *Pteropus* bats and in *Microchiroptera* (Table 1). The PREDICT_ CoV-17 sequences from *Pteropus* detected in this study clustered closely with PREDICT_ CoV-17 sequences from *Pteropus giganteus* bats in Nepal and *Pteropus lylei* bats in Thailand [36] (Fig 6, S3 Table). PREDICT_CoV-35 was found in *Microchiroptera* in bat guano farms

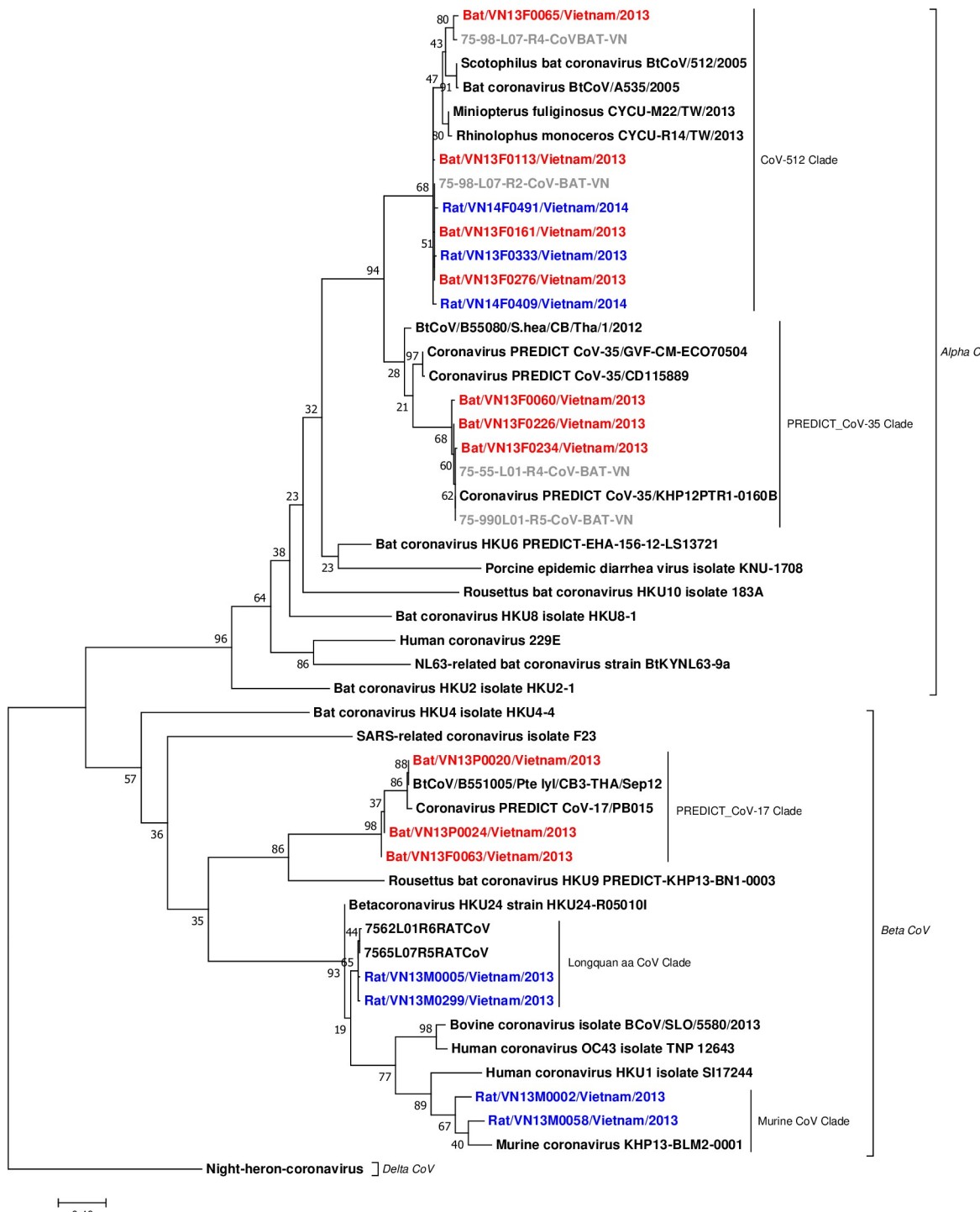

**Fig 6. Phylogenetic tree of bat and rodent coronavirus sequences detected in Viet Nam.** The analysis is based on 387 bp fragment of the *RdRp* gene using maximum likelihood with the Tamura 3-parameter model, Gamma distributed (G), and 1000 bootstrap replicates via MEGA7. The analysis included 17 sequences from this study (red from bat hosts, blue from rodent hosts), six sequences (in gray) from a previous study in Viet Nam [26], and 26 reference sequences (in black) available in the GenBank database (S3 Table). The tree was rooted by a strain of Night-heron coronavirus HKU19 (GenBank accession No. NC_016994).

and in a pteropid bat (Table 1). PREDICT_CoV-35 sequences from Viet Nam clustered with other PREDICT_CoV-35 sequences found previously in samples from hunted *Scotophilus kuhlii* bats in Cambodia (S3 Table; Dr. Lucy Keatts personal communication), and with sequences found in bats from an earlier study in the Mekong Delta region in Viet Nam (Fig 6).

Bat coronavirus 512/2005 was detected in *Microchiroptera* bat guano; and in *H. brachyura* (feces and environmental samples), *R. pruinosus* (feces barcoded), and *R. argentiventer* (barcoded environmental sample) in wildlife farms (Table 1 and S1 Table). In *Microchiroptera*, Bat coronavirus 512/2005 was frequently found in co-infection with PREDICT_CoV-35 (Table 1, S1 Table). Network analysis showed the relationships among the bat coronavirus 512/2005 sequences from the three provinces in south Viet Nam (Fig 7). We observed two main clusters and a shallow geographic structure of genetic diversity, perhaps illustrative of sampling effort but also of localized transmission and circulation of bat coronavirus 512/2005 strains in these provinces. One cluster was exclusively detected in *Microchiroptera* and mostly restricted to Dong Thap province and another cluster included sequences shared among all hosts and distributed in the three provinces (Fig 7). Parts of the network showed a star-like topology (Fig 7), typical of populations in expansion that have recently increased size. There were two sequence types that were shared among *Microchiroptera* and farmed rodents. The remaining 11 sequence types isolated from rodents on wildlife farms were not identical to those isolated from bats and were characterized by several nucleotide differences (Fig 7).

Murine coronavirus and Longquan aa coronavirus were detected in 209 and 56 field rat samples, respectively, and 26 were coinfected with both (Table 1). Two sequences of IBV were detected in rodent feces collected on two wildlife farms, one in a bamboo rat and another in a Malayan porcupine. The rodent farm interface where bat and avian coronaviruses were detected in feces were not full containment facilities and possibly had bats and birds flying and roosting overhead (Fig 2). The IBV positives were detected in fecal samples from wildlife farms that had chickens, pigs, and dogs on site.

## Discussion

### High prevalence and amplification along the supply chain for human consumption

Significant findings of this study are the high proportion of coronavirus positive wildlife (bats and rodents) and the increasing proportion of positives found along the rat trade supply chain from sub-interfaces close to the capture site (rat traders) to restaurants. The transit of multiple rat species through the supply chain, and admixing with other species and taxa at sub-interfaces along the supply chain, offers opportunities for inter- and intra-species viral exchange and recombination. Capture and transport of wildlife combined with overcrowding and close confinement of live animals in cages results in increased animal contact, likely leading to stress. While methodologically similar to rodent surveys in Zhejiang province, China (2%), Dong Thap province, Viet Nam (4.4%), and globally (0.32%), our overall proportion of coronavirus positives was much higher among field rats (34.5%) and somewhat higher among farmed rodents (6.3%) [7,26,37]. Stress, dehydration, and poor nutrition reduce animal condition and alter immune function and likely contribute to both increased shedding of viruses by infected animals, and increased susceptibility to infection of animals in the wildlife trade chain for human consumption [38].

The amplification of coronavirus along the supply chain may be seasonal as field rats were significantly more positive in the wet season. *Rattus argentiventer* generally reproduce year-round in Viet Nam, but are particularly abundant in the wet season (May through October) following the rice harvest when an abundance of food supports the population increase [39]. If

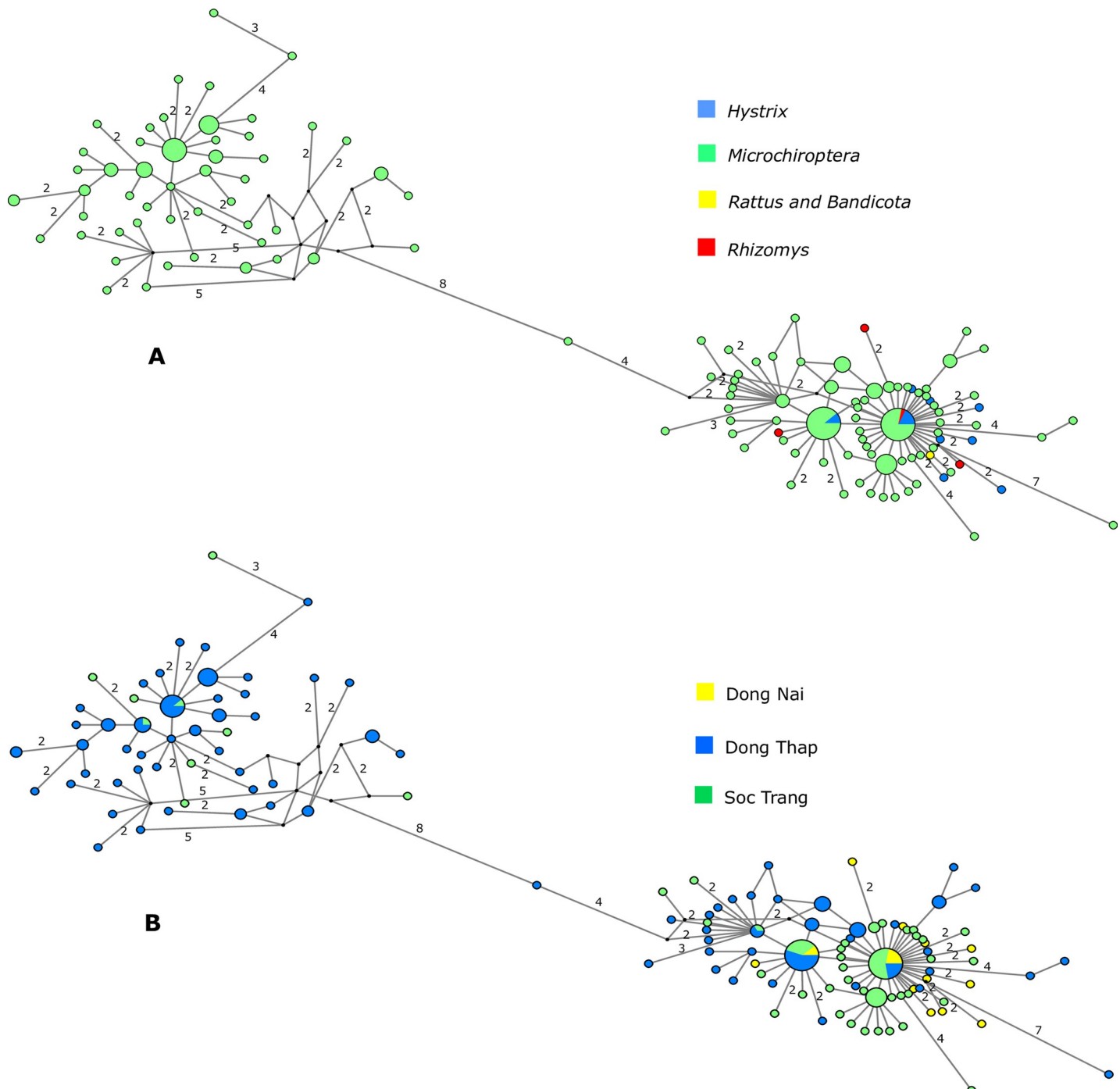

**Fig 7.** Median-joining networks of bat coronavirus 512/2005 *RdRp* sequences color-coded according to (A) host and (B) sampling location. Each circle represents a sequence type, and circle size is proportional to the number of animals sharing a sequence type. Numbers on branches indicate the number of mutations between sequence types if it was higher than one. Branches without a specified number of mutations correspond to a single mutation. Circles are colored-coded by animal host: bat (*Microchiroptera*), rodent (*Rattus* & *Bandicota*, *Rhizomys*, and *Hystrix*) and sampling location (Dong Thap (blue), Dong Nai (yellow), and Soc Trang (green)). Small black circles represent median vectors (ancestral or unsampled intermediate sequence types).

these seasonal population increases affect density dependent contact, there could be increased coronavirus prevalence and shedding in wild field rats during certain times of the year, which could then be further amplified along the trade.

Our survey was not a comprehensive multi-year evaluation of the field rat supply chain and was restricted to two provinces with this human-wildlife interface. These limitations mean we are not able to make inferences about larger spatial patterns or the inter-annual variability of coronavirus prevalence in wildlife populations found in this interface, which spans into neighboring Cambodia. Field rat carcasses were sampled immediately after they were slaughtered by traders, market vendors, or restaurant kitchen staff to optimize viral detection. Some viral cross contamination of carcasses during the butchering process may have increased the proportion of coronaviruses detected in individual animals. The degree to which cross-contamination may have elevated the proportion of coronaviruses detected in individual animals is unknown, however, this proportion accurately reflects the risk of human exposure from handling and consumption of field rats at sub-interfaces along this wild meat food chain.

From a mechanistic perspective as animals progress along the wildlife supply chain, opportunity for human contact increases, including close direct contact with traders, butchers, cooks, and consumers [40]. The combination of increased coronavirus prevalence in traded wildlife and greater opportunity for human-wildlife contact as well as intra- and inter-species contact in trade systems is likely to increase the risk of zoonotic transmission of coronaviruses in wildlife markets, restaurants, and other trade interfaces.

## Viral sharing or environmental mixing

We detected avian and bat coronaviruses in rodents raised on wildlife farms, including Malayan porcupines and bamboo rats, but we did not detect rodent-associated coronaviruses. The only previously published coronavirus testing of Malayan porcupine samples carried out in China were negative [41]. It is unclear if the Malayan porcupine samples from animals screened in this study were infected with the avian or bat viruses or if environmental contamination or mixing occurred with avian and bat guano. Chickens were present at the two sites where the IBV-positive rodents were detected, and bats fly and potentially roost overhead at most farms. 'Artificial market' studies of influenza A viruses have found cage-stacking of species on top of other species and shared water sources facilitate viral transmission [42,43]. Nevertheless, viral sharing between species and environmental contamination or mixing (i.e. bat/bird guano landing on rat feces) are two equally likely explanations for the presence of bat and avian coronaviruses detected in rodent fecal and environmental samples.

The field rats were co-infected with the Longquan aa coronavirus and the murine coronaviruses, both of which are from the Lineage A (*Embecovirus*) *Betacoronavirus* genus. Co-infections with multiple coronaviruses deserve particular attention as this co-occurrence may facilitate viral recombination leading to the emergence of new viruses [44,45].

At the very least, we conclude that rodents in the field rat and farmed rodent supply chains are being exposed to coronaviruses from rodents, bats, and birds and perhaps creating opportunities for coronavirus recombination events, which may lead to viruses that could spill over into humans [46,47]. Our findings indicate a high risk of consumer exposure even if cross-contamination due to shared butchering materials influenced the proportion of positive individuals. Repeated and more direct individual sampling of these species at these interfaces would be advantageous to determine if viral sharing was occurring versus environmental contamination of samples.

## Bat guano farms

The high proportion of positive bat feces at bat guano farms indicates the potential risk of bat guano farmers, their families, and their animals being exposed to bat coronaviruses. The overall proportion of positives (74.8%) was higher than previous studies using similar testing

methods targeting bats in Viet Nam (22%), Thailand (7.6%), Lao PDR (6.5%), and Cambodia (4.85%) [26,48,49]. In this region of Viet Nam, artificial roosts are typically erected in backyard family owned plots that incorporate a mosaic of duck, goat, or pig production and crops such as guava tress or other fruit trees and large scale kitchen gardens.

Bats have been shown to be an important evolutionary hosts of coronaviruses, including those infecting humans [7,50–53]. Both PREDICT_CoV-17 and PREDICT_CoV-35 have been detected previously in the *Pteropus* and *Microchiroptera* bats in Viet Nam, Cambodia, and Nepal, which confirms that coronaviruses are capable of infecting distantly related hosts [7]. The finding of the same virus in different bat species raises the question of whether they co-roost and/or share viruses through contact during other activities. Utilizing shared resources such as water or feeding on and around crops and fruit could lead to contact and facilitate a host jump. The presence of the same virus in bat species in multiple neighboring countries supports the suggestion by others that virus distribution coincides with their bat host distribution [7,54,55]. While there has been no testing of the pathogenicity of these bat coronaviruses in humans or animals, they are found at close contact bat-human interfaces and further characterization is needed to understand their host range and potential for spillover. Any general persecution of bats because of zoonotic viruses they may carry can actually increase the number of susceptible bats and increase transmission risk to people [56], and would interfere with the important ecosystem services that bats provide, such as controlling insect pests of rice fields [57], plant pollination, and seed dispersal.

## Capacity building and outreach

Beyond the viral findings, this work represented an important opportunity for capacity development in field, laboratory, and scientific disciplines, as well as opportunities for social engagement and education of high-risk communities on zoonotic disease threats. The consensus PCR approach for viral detection provides a cost-effective tool to detect emerging viruses in low-resource settings. Our work adds to the growing body of research demonstrating the utility of this approach to detect both known and novel viruses and co-infections in a variety of taxa, sample types, and interfaces. In Viet Nam, the direct result is an enhanced One Health surveillance capacity to detect important emerging or unknown viruses in humans, wildlife, and livestock. In the communities with which we partnered, strong engagement enabled teams to sample a wide diversity of wild animals at high-risk interfaces. Importantly, we have returned to these same communities to share the viral findings and to educate participants with an outreach program on how to live safely with bats [58].

## Conclusions

Large percentages of coronaviruses were detected in bats and rodents at sites where people have close contact and interact with wildlife including sub-interfaces along wildlife trade chains, wildlife farms, and artificial bat roosts where bat guano is collected for use as fertilizer. The high proportion of coronavirus positive samples at these human-wildlife interfaces highlights the potential for human exposure to wildlife origin coronaviruses. The observed viral amplification along the wildlife trade supply chain for human consumption, illustrated by the field rat trade in this study, likely resulted from the admixing of different species or sub-populations, and the close confinement of stressed live animals. This highlights the potential for coronavirus (and other virus) shedding and amplification along other wildlife supply chains (e.g., civets, pangolins) where similarly large numbers of animals are collected from a wide range of locations, transported, and confined. The detections of rodent, bat, and avian coronaviruses confirm concerns about productions systems and supply chains that increase contact

between wildlife and domestic species. Livestock and people living in close contact with rodents, bats, and birds shedding coronaviruses provides opportunities for intra- and inter-species transmission and potential recombination of coronaviruses.

Human behavior is facilitating the spillover of viruses, such as coronavirus, from animals to people. The wildlife trade supply chain from the field to restaurant and end consumer provides multiple opportunities for such spillover events to occur [1]. Since the SARS outbreak, broad scientific consensus exists that long term, structural changes, and wildlife trade and market closures will be required to prevent future epidemics. To minimize the public health risks of viral disease emergence from the consumption of wildlife and to safeguard livestock-based production systems, we recommend precautionary measures that restrict the killing, commercial breeding, transport, buying, selling, storage, processing and consuming of wild animals. The time has come for the global community to collectively assume responsibility through targeted wildlife trade reform. The world must also increase vigilance through building and improving detection capacity; actively conducting surveillance to detect and characterize coronaviruses in humans, wildlife, and livestock; and to inform human behaviors in order to reduce zoonotic viral transmission to humans. The more opportunities we provide for humans to come into direct contact with a multitude of wildlife species, the higher the likelihood of another spillover event. The costs of inaction are astronomically high and we must ensure that future food production and security is sustainable, just, and supports global health.

## Supporting information

**S1 Table. Summary of all testing results by genus, interface, sub-interface, sample types, sites, percentage of samples testing positive, and viral species.**
(PDF)

**S2 Table. Multivariate mixed effect logistic regression showing the association between season and sub-interface with coronavirus positives in field rats.**
(PDF)

**S3 Table. GenBank accession numbers for coronavirus sequences detected in this study and for reference sequences.**
(PDF)

**S1 Data.**
(TXT)

**S1 R Code. Code used to conduct the analysis described.**
(HTML)

## Acknowledgments

We are thankful to the government of Viet Nam, the Wildlife Conservation Society Health team for conducting field sampling, partnering laboratories for running diagnostic tests, and many other agencies for collaborations on this project. Specifically, we would like to acknowledge Le Viet Dung, Ton Ha Quoc Dung and Nguyen Van Dung (Dong Nai Province Forest Protection Department); Vo Be Hien (Dong Thap Sub-Department of Animal Health); Quach Van Tay (Soc Trang Sub-Department of Animal Health); and the late Ngo Thanh Long (Regional Animal Health Office No. 6) for his visionary leadership and commitment to this initiative in Viet Nam. The authors are indebted to the cheerful and resourceful help of Tammie O'Rourke and Dan O'Rourke and others who developed and curated the database used to

maintain PREDICT data through the Emerging Infectious Disease Information Technology Hub (EIDITH).

## Author Contributions

**Conceptualization:** Nguyen Quynh Huong, Nguyen Thi Thanh Nga, Scott I. Roberton, Nguyen Van Long, Jonna A. K. Mazet, Christine Kreuder Johnson, Tracey Goldstein, Damien O. Joly, Amanda E. Fine, Sarah H. Olson.

**Formal analysis:** Nguyen Quynh Huong, Nguyen Thi Thanh Nga, Alice Latinne, Mathieu Pruvot, Sarah H. Olson.

**Funding acquisition:** Jonna A. K. Mazet, Damien O. Joly.

**Investigation:** Nguyen Thi Thanh Nga, Le Tin Vinh Quang, Nguyen Thi Hoa, Phan Quang Minh, Nguyen Thi Diep, Nguyen Tung, Van Dang Ky, Nguyen Van Long, Tracey Goldstein, Alex Tremeau-Bravard, Victoria Ontiveros, Amanda E. Fine.

**Project administration:** Nguyen Thi Thanh Nga, Nguyen Van Long, Nguyen Van Long, Martin Gilbert, Leanne Wicker, Jonna A. K. Mazet, Christine Kreuder Johnson, Tracey Goldstein, Damien O. Joly, Amanda E. Fine, Sarah H. Olson.

**Supervision:** Nguyen Van Long, Bach Duc Luu, Nguyen Thanh Phuong, Vo Van Hung, Nguyen Thi Lan, Scott I. Roberton, Hoang Bich Thuy, Damien O. Joly, Chris Walzer, Amanda E. Fine.

**Writing – original draft:** Nguyen Quynh Huong, Nguyen Thi Thanh Nga, Alice Latinne, Mathieu Pruvot, Amanda E. Fine, Sarah H. Olson.

**Writing – review & editing:** Nguyen Quynh Huong, Nguyen Thi Thanh Nga, Nguyen Van Long, Bach Duc Luu, Alice Latinne, Mathieu Pruvot, Nguyen Thanh Phuong, Le Tin Vinh Quang, Vo Van Hung, Nguyen Thi Lan, Nguyen Thi Hoa, Phan Quang Minh, Nguyen Thi Diep, Nguyen Tung, Van Dang Ky, Scott I. Roberton, Hoang Bich Thuy, Nguyen Van Long, Martin Gilbert, Leanne Wicker, Jonna A. K. Mazet, Christine Kreuder Johnson, Tracey Goldstein, Alex Tremeau-Bravard, Victoria Ontiveros, Damien O. Joly, Chris Walzer, Amanda E. Fine, Sarah H. Olson.

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
