## [Decision Letter · Decision Letter 0]

19 Jun 2020

PONE-D-20-17798

Coronavirus testing indicates transmission risk increases along wildlife supply chains for human consumption in Viet Nam, 2013-2014

PLOS ONE

Dear Dr. Olson,

I have read your paper and found it of great interest. I therefore put it on the fast track and invited three experts reviewers who are leaders in the field to give you comments. All three are supportive and one provided detailed technical suggestions and comments that help you improve your work. You are therefore given the chance to revise your paper and address concerns raised by the three reviewers. Please send back your paper at an earliest possible time.

Thank you for submitting the paper to us and looking forward to receiveing your revised manuscript soon.

We look forward to receiving your revised manuscript.

Kind regards,

Dong-Yan Jin

Academic Editor

PLOS ONE

Journal Requirements:

2. In your Methods section, please provide additional location information of the study sites, including geographic coordinates for the data set if available.

3. In your Methods section, please provide additional information regarding the permits you obtained for the work. Please ensure you have included the full name of the authority that approved the study sites access and, if no permits were required, a brief statement explaining why.

4. We note that Figure 4 in your submission contain map images which may be copyrighted. All PLOS content is published under the Creative Commons Attribution License (CC BY 4.0), which means that the manuscript, images, and Supporting Information files will be freely available online, and any third party is permitted to access, download, copy, distribute, and use these materials in any way, even commercially, with proper attribution. For these reasons, we cannot publish previously copyrighted maps or satellite images created using proprietary data, such as Google software (Google Maps, Street View, and Earth). For more information, see our copyright guidelines: http://journals.plos.org/plosone/s/licenses-and-copyright.

4.1.    You may seek permission from the original copyright holder of Figure 4 to publish the content specifically under the CC BY 4.0 license.

4.2.    If you are unable to obtain permission from the original copyright holder to publish these figures under the CC BY 4.0 license or if the copyright holder’s requirements are incompatible with the CC BY 4.0 license, please either i) remove the figure or ii) supply a replacement figure that complies with the CC BY 4.0 license. Please check copyright information on all replacement figures and update the figure caption with source information. If applicable, please specify in the figure caption text when a figure is similar but not identical to the original image and is therefore for illustrative purposes only.

Reviewers' comments:

Reviewer's Responses to Questions

**Comments to the Author**

1. Is the manuscript technically sound, and do the data support the conclusions?

Reviewer #1: Yes

Reviewer #2: Partly

Reviewer #3: Yes

2. Has the statistical analysis been performed appropriately and rigorously? 

Reviewer #1: Yes

Reviewer #2: Yes

Reviewer #3: Yes

3. Have the authors made all data underlying the findings in their manuscript fully available?

Reviewer #1: No

Reviewer #2: Yes

Reviewer #3: Yes

4. Is the manuscript presented in an intelligible fashion and written in standard English?

Reviewer #1: Yes

Reviewer #2: Yes

Reviewer #3: Yes

5. Review Comments to the Author

Reviewer #1: Huong et al. present a coronavirus detection study in the rats and bats in Vietnam. They used PCR method for detection and amplicon sequencing for verification and characterization of the evolutionary origin of the detected coronaviruses. They found a significant prevalence of coronavirus in these animals. Interestingly, the positive rates in rat increase from farm, to markets and to restaurants. This result supports that the virus spillover risk increases along the wildlife consumption chain by human. The study is nice, and the manuscript was well written. The only things I hope could be improved are below:

In the phylogenetic tree (Figure 6) I did not see the benefit, but indeed confusing, of putting the clade name in front of each of the sequences produced by this study. E.g. Coronavirus PREDICT CoV-35/VN23F0226 and Scotophilus bat coronavirus 512 2005/PREDICT-VN13F0333. “Coronavirus PREDICT CoV-35” and “Scotophilus bat coronavirus 512 2005” are the virus that is representative of the clades where the virus strain VN23F0226 and VN13F0333 are falling into (This is what I guessed). I could not find “Coronavirus PREDICT CoV-35” nor “Scotophilus bat coronavirus 512 2005” representative strains in the tree, and I must say these are not widely-used clade names but some clades defined by one or two previous paper. I am not trying to say these clade designation are invalid, but their representative strains should be included in the tree and provided with GenBank accession number so that other researchers could follow. Using “CoV-512 Clade” etc in the vertical bar designation of the lineage are great, but then you don’t have to repeat “Scotophilus bat coronavirus 512 2005” for every strain such as VN13F0333, VN13F0161, etc. Please use some more widely used standard format for virus naming E.g. Rat/VN13F0333/Vietnam/2014. This more understandable. Putting “Scotophilus bat coronavirus 512 2005” in front of “VN13F0333” just confuse reader that it was sampled from bats, but in fact this was sampled from rats.

In the median-joining network (Figure 7), some branches did not show with the number of the mutations as specified in the legend. It would be useful if the branch are drawn in scale with the number of mutations.

Reviewer #2: This is an interesting study examining the prevalence of coronaviruses at various points in the animal/human interface in Vietnam. Although timely in the context of COVID-19, I have some issues with the sampling techniques and data presentations as below:

Major comments

• A number of terms are used to describe rodent sampling sites. ‘Trader’, ‘rodent farms’, ‘live rodent trade supply chain’, ‘Large Market’, ‘Restaurant’, ‘Live rodent trade sites’, ‘rodent wildlife farm sites’, ‘wildlife farms’, ‘rodents in the trade’ and ‘field rat trade’. On the other hand, the analysis in figure 5 clearly classifies the sites into only three types: traders, large markets and restaurants. The sampling site terminology should be clarified with straightforward definitions and the authors should stick with these definitions throughout the manuscript. I understand that the authors are trying to analyse the data according to interface or sub-interface, but it is very confusing as currently presented.

• Figure 6 shows a tree based on a 387 bp RdRp fragment. Several of the bat-derived 512 2005 and rodent-derived 512 2005 appear to have an identical sequence (see CoV-512 clade). This is rather astonishing if they are really infecting coronavirus strains from different animals. I would interpret this as a signal of cross-contamination due to amplicon carry-over in the laboratory. If the cross-contamination occurred in the field, I would expect some nucleotide differences in this fragment as I gather that the sites of rodent sampling and bat sampling are geographically distinct?? I don't think it is reasonable to conclude that bat/ avian coronaviruses can infect rodents based on the evidence presented.

• Another concern I have is cross-contamination during sampling. Faeces collected from the ground of cages (even of individually housed animals) should ideally be called environmental samples as there is no way to tell if they really originate from the animal in the cage at the time. I believe this is confirmed by the presence of IBV and bat coronaviruses in rodent farms.

• Also, if a butcher/ vendor/ chef is using the same knife to handle several animals, there would be extensive cross-contamination if one of them is positive. Would this explain the high PCR positive rates?

• The sampling is comprehensive, but also very heterogeneous in terms of type of sample collected. Why wasn’t a standard sample type applied across all sites? I am concerned whether the type of sample collected at different settings might have contributed to the apparent variation in coronavirus detection rate between settings.

Minor comments:

• The introduction could be a bit more concise.

• Line 151: Are these 28 rodent farm sites classified as ‘traders’?

• Line 174 – 175 mentions 41 + 30 = 71 sites sampled. However, line 262 and 270 mention 70 sites. Why is this?

• Figure 4: Bars representing Sciuridae and R. argentiventer are not presented in this figure. Why are they in the legend?

• Figure 5: Could add between-group comparisons by chi-square to this figure.

Reviewer #3: In this manuscript, the authors have performed a surveillance study on the presence of the coronaviruses in the wildlife and the wildlife-human interfaces in Viet Nam. They detected the coronaviruses by consensus PCR and discovered the infected cases were increased along the supply chain. Although this study is performed in 2013-2014, its findings have a very good insight into the current COVID-19 outbreak, especially showing the potential risk of spreading the coronaviruses along the wild animal trade supply chain. Their findings can help to dissect how coronaviruses can spread from wild animals to humans by social activities. In view of this, I support to publish this manuscript in PLoS-One.

6. PLOS authors have the option to publish the peer review history of their article (what does this mean?). If published, this will include your full peer review and any attached files.

Reviewer #1: No

Reviewer #2: No

Reviewer #3: No

---

## [Author Response · Author response to Decision Letter 0]

17 Jul 2020

Journal Requirements:

Response: Thank you for these reminders. We reviewed and applied these requirements where there were discrepancies, in particular removing the list of key words and the funding information from the acknowledgment section. In addition we re-assigned the header styles, which had somehow disappeared from the formatting, and revised the order of affiliation for one of the co-authors. 

2. In your Methods section, please provide additional location information of the study sites, including geographic coordinates for the data set if available.

Response: Our analysis presented location data at the province level and a spatial map of those provinces are provided in Figure 4. In response to this request we included the district name as well as the latitude and longitude rounded to two significant digits for each test result. The revised S1 Data table available at (pending DOI processing): https://doi.org/10.5061/dryad.7h44j0zrj OR https://datadryad.org/stash/share/pk3wVUxFNzTuCYZ9t8haKRPmx7V8YhTDBuHpG8JJ9kU.

3. In your Methods section, please provide additional information regarding the permits you obtained for the work. Please ensure you have included the full name of the authority that approved the study sites access and, if no permits were required, a brief statement explaining why.

Response: Thank you for this comment. The ‘permitting’ process in Viet Nam is perhaps a bit unusual to those not conducting wildlife research in the country. We have added some additional context to the relevant statement in the manuscript’s method section, which now reads, ‘The study and sampling activities for specified dates and locations were approved by the Department of Animal Health of the Ministry of Agriculture and Rural Development and animal sampling protocols were reviewed by the Institutional Animal Care and Use Committee at the University of California at Davis (protocol number 16048)’. These constitute the full names of the Vietnamese authorities that approved access to the study sites. The study approval with the Department of Animal Health (DAH) was confirmed in an Memorandum of Understanding dated September 2011, and a Work Plan Agreement dated August 16, 2012. The Department of Animal Health used internal communication channels to confirm their approval for the activities and directed the provincial level Department of Animal Health (Soc Trang Province subDAH, Dong Thap Province subDAH, and Dong Nai Province subDAH) to support the sampling at the study sites. No permit numbers were provided by the Vietnamese authorities. 

4. We note that Figure 4 in your submission contain map images which may be copyrighted. 

Response: We switched our vector layer to a source that is CC BY 4.0 compliant. The source of the Viet Nam province level vector layer is now geoBoundaries v. 3.0.0 (https://www.geoboundaries.org; Runfola et al. 2020) and Open Development Mekong (ODM; https://vietnam.opendevelopmentmekong.net). Note the geoBoundaries record of the latest ODM source license is not yet updated to reflect a recent change by ODM to CC BY 4.0 as indicated by the following ticket: https://github.com/wmgeolab/gbRelease/issues/53. The attributions to these sources have been updated in the methods section and in the Figure 4 legend.

Reviewers' comments:

1. Is the manuscript technically sound, and do the data support the conclusions?

Reviewer #1: Yes

Reviewer #2: Partly

Reviewer #3: Yes

2. Has the statistical analysis been performed appropriately and rigorously? 

Reviewer #1: Yes

Reviewer #2: Yes

Reviewer #3: Yes

3. Have the authors made all data underlying the findings in their manuscript fully available?

Reviewer #1: No

Reviewer #2: Yes

Reviewer #3: Yes

Response: In our supporting information we provide the following statement on S1 Data: ‘Data required for all analysis and metadata for each parameter, now including site location information for each test, is available at (pending DOI processing): https://doi.org/10.5061/dryad.7h44j0zrj OR https://datadryad.org/stash/share/pk3wVUxFNzTuCYZ9t8haKRPmx7V8YhTDBuHpG8JJ9kU

4. Is the manuscript presented in an intelligible fashion and written in standard English?

PLOS ONE does not copy edit accepted manuscripts, so the language in submitted articles must be clear, correct, and unambiguous. Any typographical or grammatical errors should be corrected at revision, so please note any specific errors here.

Reviewer #1: Yes

Reviewer #2: Yes

Reviewer #3: Yes 

5. Review Comments to the Author

Reviewer #1: Huong et al. present a coronavirus detection study in the rats and bats in Vietnam. They used PCR method for detection and amplicon sequencing for verification and characterization of the evolutionary origin of the detected coronaviruses. They found a significant prevalence of coronavirus in these animals. Interestingly, the positive rates in rat increase from farm, to markets and to restaurants. This result supports that the virus spillover risk increases along the wildlife consumption chain by human. The study is nice, and the manuscript was well written. 

Response: Thank you for this support.

The only things I hope could be improved are below:

In the phylogenetic tree (Figure 6) I did not see the benefit, but indeed confusing, of putting the clade name in front of each of the sequences produced by this study. E.g. Coronavirus PREDICT CoV-35/VN23F0226 and Scotophilus bat coronavirus 512 2005/PREDICT-VN13F0333. “Coronavirus PREDICT CoV-35” and “Scotophilus bat coronavirus 512 2005” are the virus that is representative of the clades where the virus strain VN23F0226 and VN13F0333 are falling into (This is what I guessed). I could not find “Coronavirus PREDICT CoV-35” nor “Scotophilus bat coronavirus 512 2005” representative strains in the tree, and I must say these are not widely-used clade names but some clades defined by one or two previous paper. I am not trying to say these clade designation are invalid, but their representative strains should be included in the tree and provided with GenBank accession number so that other researchers could follow. Using “CoV-512 Clade” etc in the vertical bar designation of the lineage are great, but then you don’t have to repeat “Scotophilus bat coronavirus 512 2005” for every strain such as VN13F0333, VN13F0161, etc. Please use some more widely used standard format for virus naming E.g. Rat/VN13F0333/Vietnam/2014. This more understandable. Putting “Scotophilus bat coronavirus 512 2005” in front of “VN13F0333” just confuse reader that it was sampled from bats, but in fact this was sampled from rats.

Response: We renamed all sequences included in our phylogenetic tree following the Reviewer’s suggestion (e.g. Rat/VN13F0333/Vietnam/2013). PREDICT_CoV-35 and PREDICT_CoV-17 were first described in Anthony et al. 2017 and we used the same clade names in this study. Representative sequences for PREDICT_CoV-35 and PREDICT_CoV-17 were included in the tree (PREDICT_CoV-17: KX284941, PREDICT_CoV-35 = KX284991 and KX285074). We added another sequence (DQ648858) to our phylogenetic tree as a representative sequence of the Scotophilus bat coronavirus 512 clade.

In the median-joining network (Figure 7), some branches did not show with the number of the mutations as specified in the legend. It would be useful if the branch are drawn in scale with the number of mutations.

Response: Number of mutations were indicated above branches only when this number was higher than one. All branches without a specified number of mutations correspond to a single mutation as it is now explained in the figure legend. When possible, we re-drew the length of some branches to scale to make it proportional to the number of mutations. However this was not always possible due to the complexity of the network.

Reviewer #2: This is an interesting study examining the prevalence of coronaviruses at various points in the animal/human interface in Vietnam. Although timely in the context of COVID-19, I have some issues with the sampling techniques and data presentations as below:

Major comments

• A number of terms are used to describe rodent sampling sites. ‘Trader’, ‘rodent farms’, ‘live rodent trade supply chain’, ‘Large Market’, ‘Restaurant’, ‘Live rodent trade sites’, ‘rodent wildlife farm sites’, ‘wildlife farms’, ‘rodents in the trade’ and ‘field rat trade’. On the other hand, the analysis in figure 5 clearly classifies the sites into only three types: traders, large markets and restaurants. The sampling site terminology should be clarified with straightforward definitions and the authors should stick with these definitions throughout the manuscript. I understand that the authors are trying to analyse the data according to interface or sub-interface, but it is very confusing as currently presented.

Response: We strived to consistently use a minimal and descriptive set of terms but as Reviewer 2 points out, there was still considerable room for improvement. Throughout we have revised the phrasing to create more consistency and to further clarify the distinction between the field rat trade associated sub-interfaces and the wildlife farm rodents. Figure 4 & 5 are now redesigned to better align with the interface listed in Table 1, which was intentionally configured to help readers understand the main interfaces and sub-interfaces. Figure 5 focuses on the field rat trade sub-interfaces where sampling was standardized and consistent across sites, and therefore does not include data from other interfaces.

• Figure 6 shows a tree based on a 387 bp RdRp fragment. Several of the bat-derived 512 2005 and rodent-derived 512 2005 appear to have an identical sequence (see CoV-512 clade). This is rather astonishing if they are really infecting coronavirus strains from different animals. I would interpret this as a signal of cross-contamination due to amplicon carry-over in the laboratory. If the cross-contamination occurred in the field, I would expect some nucleotide differences in this fragment as I gather that the sites of rodent sampling and bat sampling are geographically distinct?? I don't think it is reasonable to conclude that bat/ avian coronaviruses can infect rodents based on the evidence presented.

Response: We identified 13 CoV 512 sequence types in rodents. Among these 13 sequence types, only two of them were identical to sequence types isolated from bats. The remaining 11 sequence types isolated from rodents were not identical to those isolated from bats and were characterized by several nucleotide differences; this is clearly visible in the phylogenetic network in Fig 7. We can therefore conclude that rodents were infected by slightly divergent but closely related CoV 512 strains and that this was not the result of a lab contamination as most of the rodent sequence types (11/13) were never detected in our bat samples. All rodent sequence types were included in the phylogenetic network (Fig 7) and colored in red, blue, and yellow (Fig 7) but only 3 of them were included in the phylogenetic tree (Fig 6) as representative sequences of the whole dataset. To address this well-stated concern for other readers, in our revised submission we have re-run our phylogenetic tree using more divergent rodent CoV 512 sequence representatives so it is easier to see in the phylogenetic tree that sequence types from rodents and bats are not identical. 

• Another concern I have is cross-contamination during sampling. Faeces collected from the ground of cages (even of individually housed animals) should ideally be called environmental samples as there is no way to tell if they really originate from the animal in the cage at the time. I believe this is confirmed by the presence of IBV and bat coronaviruses in rodent farms.

Response: Additional details were included to clarify the criteria used to distinguish between what were called “environmental samples” and feces or urine collected non-invasively from animals at sites. Feces and urine samples voided from animals observed in cages (individual animals in cages) or in the case of the bat sampling, visible above in bat roots, these were identified as feces and urine. Swabs of feces piles in cages or surfaces of the animal cages on wildlife farms were identified as environmental samples. 

• Also, if a butcher/ vendor/ chef is using the same knife to handle several animals, there would be extensive cross-contamination if one of them is positive. Would this explain the high PCR positive rates?

Response: Yes, it is possible that as the field rats are slaughtered there is cross contamination of individual animals through knives used or contaminated surfaces. Sample collection from individual animals, however, involved using individual sterile swabs and sterile sample collection tools, to prevent any cross contamination during sample collection. The proportion of viruses we identified was the proportion identified in individual animals destined for human consumption so the proportion of positives reflects the proportion of exposure of the trader, market vendor, restaurant butcher, or end consumer to coronaviruses at those points of contact or interfaces between wildlife and humans. The high rates of positives and increase in positive rate along the field rat trade chain was across all sites and periods of sample collection. To make this situation clearer to readers we’ve added the following text to the discussion: ‘Our findings indicate a high risk of consumer exposure even if cross-contamination due to shared butchering materials influenced the proportion of positive individuals (see Methods).’

• The sampling is comprehensive, but also very heterogeneous in terms of type of sample collected. Why wasn’t a standard sample type applied across all sites? I am concerned whether the type of sample collected at different settings might have contributed to the apparent variation in coronavirus detection rate between settings.

Response: We agree that this is a caveat in the study, and we have transparently highlighted this in the description of the Material and Methods and the results (including in the shared code). We agree that ideally, the same sample types would have been collected consistently from all settings/interfaces, however, the different settings/interfaces did not allow identical access to all sample types. These inconsistencies are the reason why we did not conduct analysis across the entire dataset, but instead focused analysis where it could be supported by the data (i.e. we only compared what was comparable).

Minor comments:

• The introduction could be a bit more concise.

Response: We tightened up the introduction by removing some lengthier text while maintaining the relevant background and content as these interfaces are relatively unique to this region and we believe the readership will benefit from more detailed descriptions.

• Line 151: Are these 28 rodent farm sites classified as ‘traders’?

Response: No the rodent farms are not classified as traders. We have updated the text throughout to clarify what is the field rat trade chain with vendors as the first sub-interface in the trade chain and indicated that the rodents raised in captivity on wildlife farms (porcupines and bamboo rats) are a separate interface. 

• Line 174 – 175 mentions 41 + 30 = 71 sites sampled. However, line 262 and 270 mention 70 sites. Why is this?

Response: The bat pagoda site was the only site that was sampled in both the dry and the wet season. The lines in question clarify this with the following now revised statement ‘Given the distances between sites, 69 sites were sampled once except the bat pagoda natural roost in Soc Trang province, which was visited three times and sampled in both seasons’. Also of note, in the process of reviewing this comment we discovered and fixed a basic calculation error in the last row of Table 1. 

• Figure 4: Bars representing Sciuridae and R. argentiventer are not presented in this figure. Why are they in the legend?

Response: We struggled with a way to represent Sciuridae and Rattus argentiventer in this figure because they were only sampled one time apiece from wildlife farms (see Table 1). They are present in the figure (if one zooms in they are visible as a red/purple line at the bottom of Dong Nai barplot) and we added a statement to the figure legend because of this concern.

• Figure 5: Could add between-group comparisons by chi-square to this figure.

Response: We vacillated between reporting univariate (i.e. chi-square statistics) versus multivariate model statistics for the amplification finding. In lieu of adding chi-square statistics to this figure we point the reviewer to the multivariate model (S2 Table) that allowed us to adjust for sites as random effects, and which also show the between group findings are significantly different.

Reviewer #3: In this manuscript, the authors have performed a surveillance study on the presence of the coronaviruses in the wildlife and the wildlife-human interfaces in Viet Nam. They detected the coronaviruses by consensus PCR and discovered the infected cases were increased along the supply chain. Although this study is performed in 2013-2014, its findings have a very good insight into the current COVID-19 outbreak, especially showing the potential risk of spreading the coronaviruses along the wild animal trade supply chain. Their findings can help to dissect how coronaviruses can spread from wild animals to humans by social activities. In view of this, I support to publish this manuscript in PLoS-One.

Response: Thank you for this support.

---

## [Decision Letter · Decision Letter 1]

22 Jul 2020

Coronavirus testing indicates transmission risk increases along wildlife supply chains for human consumption in Viet Nam, 2013-2014

PONE-D-20-17798R1

Dear Dr. Olson,

I have put your manuscript in the fast track in both rounds of review. Your revised paper has now been reviewed by one original reviewer and he promptly recommended acceptance of your work for publication, We’re therefore pleased to inform you that your manuscript has been judged scientifically suitable for publication and will be formally accepted for publication once it meets all outstanding technical requirements. Congratulations!

Kind regards,

Dong-Yan Jin

Academic Editor

PLOS ONE

Reviewers' comments:

Reviewer's Responses to Questions

**Comments to the Author**

1. If the authors have adequately addressed your comments raised in a previous round of review and you feel that this manuscript is now acceptable for publication, you may indicate that here to bypass the “Comments to the Author” section, enter your conflict of interest statement in the “Confidential to Editor” section, and submit your "Accept" recommendation.

Reviewer #2: All comments have been addressed

2. Is the manuscript technically sound, and do the data support the conclusions?

Reviewer #2: Yes

3. Has the statistical analysis been performed appropriately and rigorously? 

Reviewer #2: Yes

4. Have the authors made all data underlying the findings in their manuscript fully available?

Reviewer #2: Yes

5. Is the manuscript presented in an intelligible fashion and written in standard English?

Reviewer #2: Yes

6. Review Comments to the Author

Reviewer #2: Thanks for addressing the comments. The manuscript has been improved considerably with the clarifications of the terminology.

7. PLOS authors have the option to publish the peer review history of their article (what does this mean?). If published, this will include your full peer review and any attached files.

Reviewer #2: **Yes: **Siddharth Sridhar

---

## [Editor Report · Acceptance letter]

28 Jul 2020

PONE-D-20-17798R1 

Coronavirus testing indicates transmission risk increases along wildlife supply chains for human consumption in Viet Nam, 2013-2014 

Dear Dr. Olson:

I'm pleased to inform you that your manuscript has been deemed suitable for publication in PLOS ONE. Congratulations! Your manuscript is now with our production department. 

Kind regards, 

on behalf of

Professor Dong-Yan Jin 

Academic Editor

PLOS ONE